# Influencing factors of HCV testing willingness among men who have sex with men in China: A structural equation modeling analysis

Jiayan Li[1], Jing He[1], Guohui Yang[1], Zhen Cao[1,2], Xiaoni Zhong[1]*

**1** School of Public Health, Chongqing Medical University, Chongqing, China, **2** Chongqing Yubei District Center for Disease Control and Prevention, Chongqing, China

* zhongxiaoni@cqmu.edu.cn

## Abstract

### Background

Men who have sex with men (MSM) are a population with greater likelihood of the hepatitis C virus (HCV). Early detection of HCV status is beneficial to therapy and prevention of the further transmit of HCV. The testing rate of HCV in MSM is low, so it is important to investigate the factors that influence testing willingness in order to increase the testing rate.

### Objectives

Based on the health belief model (HBM), this study investigated the influencing factors of HCV testing willingness among MSM, and provided a basis for promoting HCV testing among MSM.

### Methods

Non-probability sampling was employed to collect samples, and electronic questionnaires were used to perform cross-sectional surveys on the samples, including socio-demographic characteristics, sexual behavior characteristics, HCV testing willingness and HBM scale. The data was evaluated with a structural equation model (SEM).

### Results

Result of the 857 MSM, 55.7% had ever undergone HCV testing, 90.9% were willing to undergo HCV testing in the future, and 71.7% were anticipated to undergo HCV testing within the next six months. The SEM's findings demonstrated that behavioral intention was positively impacted by self-efficacy ($\beta = 0.482$, $p < 0.001$) and perceived susceptibility ($\beta = 0.312$, $p < 0.001$), while behavioral intention was negatively impacted by perceived barriers ($\beta = -0.254$, $p < 0.001$). Furthermore, behavioral intention was indirectly impacted by perceived benefits ($\beta = 0.309$, $p < 0.001$) and perceived barriers ($\beta = -0.139$, $p < 0.001$), with self-efficacy acting as a mediating factor.

**Data availability statement:** All relevant data are within the manuscript and its Supporting Information files.

**Funding:** This work was supported by the National Key Project for Infectious Diseases of the Ministry of Science and Technology of China (2018ZX10721102-005). The funders had no role in study design, data collection and analysis, decision to publish, or preparation of the manuscript.

**Competing interests:** The authors have declared that no competing interests exist.

## Conclusion

Self-efficacy, perceived susceptibility, benefits, and barriers predict behavioral intention. These findings can inform the development of methods to increase MSM willingness to identify HCV.

## 1 Introduction

Global public health is at risk from hepatitis C virus (HCV) infection. Hepatitis brought on by HCV infection can be acute or chronic, mild or life-long, can cause liver cancer and cirrhosis, among other serious complications [1]. According to estimates from the World Health Organization (WHO), there are 58 million chronic HCV infections globally [2]. With an estimated 9.8 million chronic HCV infections, China is among the nations with the highest number of HCV infections [3]. In order to eradicate HCV as a danger to global public health by 2030, the WHO suggested in 2016 that HCV incidence be reduced by 90% and HCV-associated death by 65% [4]. HCV is a mostly occult infection. Although full-course standardized hepatitis C antiviral drugs can cure more than 95% of hepatitis C patients, the accessibility of diagnosis and treatment is very low [5,6]. In 2016, only 18% of HCV patients in China were diagnosed [7]. Early screening and diagnosis remain the primary barriers to eradicating HCV [6].

Men who have sex with men (MSM) are a high - incidence population of HCV [8–10], particularly those people living with HIV [11,12]. According to a worldwide systematic analysis, the combined prevalence of HCV in MSM was estimated to be 3.4% (95%CI: 2.8–4.0%), with 6.3% (95%CI: 5.3–7.5%) of HCV prevalence in HIV-positive populations [13]. HCV prevalence among MSM in China was estimated to be 0.67% (95%CI: 0.51–0.86%) [14]. However, the active test of MSM is poor, and the HCV testing rate is generally low. According to a 2017 nationwide survey, just 41% of Chinese MSM had ever undergone HCV testing [15].

HCV testing can diagnose and treat person with HCV earlier, so as to prevent or delay the development of hepatitis-related liver diseases and prevent the further transmit of HCV. Few studies have explored the variables that influence HCV testing in MSM, and past research has relied on univariate analysis without considering interrelationships. Therefore, we aimed to explore the willingness of MSM to accept HCV testing and construct a theoretical model based on the Health Belief Model (HBM) to explore the influencing factors of HCV testing in MSM population. To clarify the degree and mode of influence of each core element (such as perceived susceptibility, perceived severity, perceived benefits and perceived barriers) in the model on the willingness to test for HCV in MSM in China, and provide evidence for the promotion of HCV testing in MSM population.

## 2 Methods

### 2.1 Participants and recruitment

From December 1–13, 2023, MSM volunteers in China were sought via the non-probability sampling method, making use of online platforms such as WeChat official accounts frequently used by MSM, peer introductions, non-governmental organization (NGO) collaboration, and the "snowballing" of core members. The following were the requirements for MSM's inclusion: (1) have had sex with men within the last 12 months; (2) be able to fill out the questionnaire independently; (3) between the ages of 18 and 65. Exclusion criteria include: (1) taking fewer than three minutes to complete the questionnaire (after testing, we assessed that participants should have completed the questionnaire in at least 3 min.); (2) having logical problems in the questionnaires (the logical test questions were answered incorrectly, or the answers to

some questions in the questionnaire were contradictory). The questionnaire was set up with screening questions to identify the key population. An anonymous cross-sectional online poll was used to perform this investigation. Participants that completed the questionnaire and passed the review earned a reward of 5 RMB (about $0.69), which was informed to them prior to obtaining informed consent.

## 2.2 Measurements

**2.2.1 Personal characteristics questionnaire.** The collected information consisted of three parts: (1) sociodemographic characteristics (such as age, ethnicity, household registration, education level, employment status, marital status, etc.); (2) sexual behavioral characteristics (e.g., number of male sex partners, condom use, presence of female sex partners, commercial sex, stimulant use, etc.) in the past 6 months; (3) other questions (we collected self-reported data on whether participants were diagnosed with any sexually transmitted infection in the last six months, and ever tested for HCV in the past, etc.)

**2.2.2 HBM theoretical model and scale.** HBM is a valuable theoretical model for explaining health-related behaviors using social psychology approaches, highlighting the role of perception in decision-making. The factors that determine behavioral intention are perceived susceptibility, perceived severity, perceived benefits, perceived barriers, and self-efficacy [16,17]. In addition, studies have shown that perceived benefits or barriers can indirectly affect outcome variable through self-efficacy [18–21]. Based on the above theories, the initial SEM of HBM is shown in Fig 1.

The 23-item scale has five categories (Cronbach's α = 0.816) and was based on the HBM and previous research [22–24]. MSM were asked about their views on HCV testing, which helped to assess factors influencing HCV testing among MSM. The Likert five-point method was used to assess the scale (Table 1).

Perceived susceptibility is a person's evaluation of their likelihood of HCV. The perceived susceptibility scale was composed of three questions (Cronbach's α = 0.714), with a greater score suggesting a higher probability of HCV. Perceived severity is the perception of the severity and probable consequences of HCV infection. It included three questions (Cronbach's α = 0.812), and the higher the score, the more severe hepatitis C was considered. Perceived benefits relate to a person's view of the physical and psychological benefits of HCV testing. There were six items in this section (Cronbach's α = 0.896), and higher scores showed more perceived benefits from HCV testing. Perceived barriers refer to an assessment of the difficulty and cost of HCV testing. This section was measured using four questions (Cronbach's

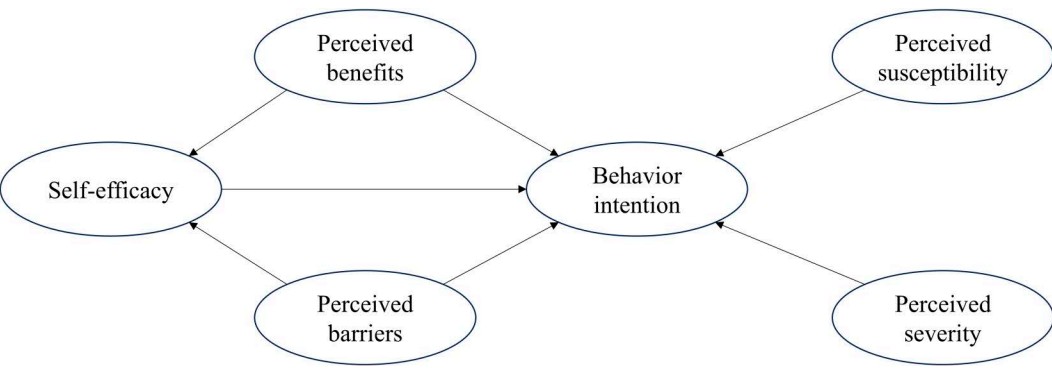

**Fig 1. The initial structural equation model of HBM.**

**Table 1. The HBM scale and value assignment.**

| Constructs | Value Assignment |
|---|---|
| Perceived susceptibility (PSU) | |
| PSU1 How likely do you think you have hepatitis C? | 1 = very small, 2 = small, 3 = average, 4 = large, 5 = very large |
| PSU2 How likely do you think men who have sex with men are to contract hepatitis C? | |
| PSU3 I am more susceptible to infection than others. | 1 = strongly disagree, 2 = disagree, 3 = neutral, 4 = agree, 5 = strongly agree |
| Perceived severity (PSE) | |
| PSE1 Hepatitis C poses a serious threat to human health. | 1 = strongly disagree, 2 = disagree, 3 = neutral, 4 = agree, 5 = strongly agree |
| PSE2 If I have hepatitis C, it will have a serious impact on my health. | |
| PSE3 If I have hepatitis C, it will have a great impact on daily life. | |
| Perceived benefits (PBE) | |
| PBE1 Detection can detect HCV infection earlier, and the treatment effect is better. | 1 = strongly disagree, 2 = disagree, 3 = neutral, 4 = agree, 5 = strongly agree |
| PBE2 Getting tested for HCV gives me peace of mind. | |
| PBE3 HCV testing can reduce my concerns. | |
| PBE4 HCV testing is a guarantee for my health. | |
| PBE5 HCV testing can prevent the transmission of potential infections to sexual partners or family members. | |
| PBE6 Acceptance of HCV testing makes me safer and keeps me away from hepatitis/cirrhosis/liver cancer. | |
| Perceived barriers (PBA) | |
| PBA1 I don't have enough access to HCV testing information. | 1 = strongly disagree, 2 = disagree, 3 = neutral, 4 = agree, 5 = strongly agree |
| PBA2 HCV testing is uncomfortable or painful. | |
| PBA3 I don't have time to test it. | |
| PBA4 The cost of HCV testing is high. | |
| Self-efficient (SE) | |
| SE1 HCV testing is convenient. | 1 = strongly disagree, 2 = disagree, 3 = neutral, 4 = agree, 5 = strongly agree |
| SE2 It's easy for me to get tested. | |
| SE3 If I want to, I am confident of successfully completing HCV testing. | |
| SE4 Despite the setbacks, I will continue to fight to get tested. | |
| SE5 When I decide to test, nothing can stop me. | |
| Behavior intention (BI) | |
| BI1 Whether you are willing to take the initiative to test for HCV. | 1 = very reluctant, 2 = reluctant, 3 = neutral, 4 = willing, 5 = very willing |
| BI2 The likelihood that you will take the initiative to test for HCV in the next 6 months. | 1 = very small, 2 = small, 3 = average, 4 = large, 5 = very large |

α = 0.785), with a higher score suggesting greater perceived barriers. Self-efficacy relates to confidence in one's capacity to finish HCV testing, this part included five items (Cronbach's α = 0.847), with higher scores indicating greater confidence in the capacity to successfully test for HCV. Behavioral intention to test for HCV was measured by willingness and likelihood in the next six months, with higher scores indicating higher intention to test for HCV.

## 2.3 Statistical analysis

In this study, descriptive analysis was performed using SPSS 26.0. Categorical data were described using frequencies and percentages. Mplus8.3 was used for structural equation modeling, and weighted least squares with mean and variance adjusted (WLSMV) was used to estimate the parameters. Composite reliability (CR) values above 0.70 indicate strong internal consistency. Convergent validity was measured using average variance extraction (AVE), with

scores above 0.50 indicating good convergence. To assess discriminant validity, the square root of the AVE value and correlation coefficient were used. Good discriminant validity was shown when the square root of each common factor's AVE value exceeded the standardized correlation coefficient with other components. $R^2$ was employed to represent the degree to which the independent variable was interpreted in terms of the dependent variable.

Model fit was evaluated using the chi-squared and degree of freedom ratio ($\chi^2$/df), comparative fit index (CFI), Tucker Lewis index (TLI), standardized root mean square residual (SRMR), and root mean square error of approximation (RMSEA) [25]. The model fitted well when $\chi^2$/df < 5, CFI > 0.90, TLI > 0.90, SRMR < 0.08 and RMSEA < 0.08. A *p*-value of < 0.05 was judged statistically significant. The Bootstrap method was used to test the 95% confidence interval [26].

### 2.4 Ethical approval

This study was approved by the Ethics Committee of Chongqing Medical University (2019001, 28 May 2019). The participants were informed consent. Before starting the survey, the study objectives were informed to the participants. Respondents were adequately informed that their information would be confidential and no identifiable information would be disclosed. Moreover, it is clearly stated that the participants can withdraw from the survey at any point. We obtained implied consent. We have provided a detailed description of the above content on the first page of the questionnaire. If participants agree to take part in the study, they can proceed to the next page on their own to fill out the questionnaire.

## 3 Results

### 3.1 Sociodemographic characteristics

The study recruited 1215 participants in total, of whom 358 (29.47%) were eliminated for the reasons listed below: 8 were age < 18 or > 65, 124 were completed questionnaires in less than 3 minutes, 226 had problems with logical check, there were finally 857 qualified respondents. 67.9% of whom were 18–26 years old, 98.9% were Han nationalities, 74.2% were urban, 64.3% College and above, 77.6% were employed, 83.9% were unmarried, and 50.1% had a monthly disposable income of 5000–10000 RMB. In terms of sexual behavior characteristics in the past 6 months, 65.3% of MSM had only one sex partner, 65.9% of MSM used condoms each time they had anal intercourse with a male partner, 31.2% used it sometimes or occasionally, 20.5% said they had a female sex partner, 12.6% had had commercial sex. In addition, 11.7% had been diagnosed with an STD in the last six months. 55.7% had been ever tested for HCV, 90.9% of MSM said they were willing or very willing to be tested for HCV in the future, and 71.7% of MSM said they were high or very likely to be tested for HCV in the next six months. (Table 2)

### 3.2 Measurement Model

Table 3 shows the factor loading, CR, and AVE for each construct, and discriminant validity. All factor loads were over 0.6, indicating acceptable levels and statistical significance (*p* < 0.001). The data have strong internal consistency and convergence validity (CR > 0.7, AVE > 0.5). Additionally, the square root of AVE was higher than other correlation coefficients, showing adequate discriminant validity. The SEM was utilized for confirmatory factor analysis. The model fit indices were as follows: $\chi^2$/df = 4.621, CFI = 0.954, TLI = 0.946, SRMR = 0.048, RMSEA = 0.065. The fit indices reached the specified levels, suggesting the model was acceptable.

Table 2. Characteristics of the study sample (N = 857).

| Variables | Classification | N | % |
|---|---|---|---|
| Age | 18-26 | 582 | 67.9 |
| | 27-35 | 242 | 28.2 |
| | >35 | 33 | 3.9 |
| Ethnicity | Han nationality | 848 | 98.9 |
| | Other ethnic minorities | 9 | 1.1 |
| Household registration | Urban areas | 636 | 74.2 |
| | Rural areas | 221 | 25.8 |
| Educational level | Junior high and below | 12 | 1.4 |
| | High school | 67 | 7.8 |
| | Junior college | 227 | 26.5 |
| | College and above | 551 | 64.3 |
| Employment status | Employed | 665 | 77.6 |
| | Retired or unemployed | 27 | 3.1 |
| | Students | 165 | 19.3 |
| Marital status | Unmarried | 719 | 83.9 |
| | Married | 122 | 14.2 |
| | Divorced/widowed | 16 | 1.9 |
| Personal monthly income | 3000 RMB or less | 129 | 15 |
| | 3001 ~ 5000 RMB | 199 | 23.2 |
| | 5001 ~ 10000 RMB | 429 | 50.1 |
| | 10000 RMB or more | 100 | 11.7 |
| The number of male partners in the past 6 months. | 0 | 5 | 0.6 |
| | 1 | 560 | 65.3 |
| | ≥2 | 292 | 34.1 |
| Condom use when having anal sex with a male partner in the past 6 months. | No anal sex | 5 | 0.6 |
| | Every time | 560 | 65.3 |
| | Sometimes | 267 | 31.2 |
| | Rarely | 25 | 2.9 |
| Have you had a female sexual partner in the last 6 months? | Yes | 176 | 20.5 |
| | No | 681 | 79.5 |
| Have you had commercial sex in the last 6 months? | Yes | 108 | 12.6 |
| | No | 749 | 87.4 |
| Have you used stimulants such as rush in the past 6 months? | Yes | 241 | 28.1 |
| | No | 616 | 71.9 |
| Have you ever undergone any traumatic procedures (e.g., surgery, piercing, tattoo, piercing, etc.)? | Yes | 393 | 45.9 |
| | No | 464 | 54.1 |
| Have you ever been diagnosed with any sexually transmitted infection in the last 6 months? | Yes | 100 | 11.7 |
| | No | 757 | 88.3 |
| Have you ever been tested for HCV? | Yes | 477 | 55.7 |
| | No | 380 | 44.3 |
| Would you be willing to actively test for hepatitis C virus in the future? | Very willing | 485 | 56.6 |
| | Willing | 294 | 34.3 |
| | Neutral | 67 | 7.8 |
| | Reluctant | 10 | 1.2 |
| | Very reluctant | 1 | 0.1 |

*(Continued)*

**Table 2.** (Continued)

| Variables | Classification | N | % |
|---|---|---|---|
| The likelihood that you will take the initiative to test for HCV in the next 6 months. | Very large | 297 | 34.7 |
| | Large | 317 | 37.0 |
| | Average | 197 | 23.0 |
| | Small | 36 | 4.2 |
| | Very small | 10 | 1.2 |

**Table 3. Factor loading, convergent validity, reliability analysis, and discriminant validity.**

| Constructs | Factor Loading | CR (>0.7) | AVE (>0.5) | PSU | PSE | PBE | PBA | SE | BI |
|---|---|---|---|---|---|---|---|---|---|
| PSU | 0.643-0.840 | 0.758 | 0.515 | **0.718** | | | | | |
| PSE | 0.778-0.865 | 0.863 | 0.677 | 0.208[***] | **0.823** | | | | |
| PBE | 0.801-0.844 | 0.926 | 0.676 | 0.158[***] | 0.814[***] | **0.822** | | | |
| PBA | 0.613-0.831 | 0.819 | 0.533 | 0.028 | -0.148[***] | -0.215[***] | **0.730** | | |
| SE | 0.682-0.841 | 0.881 | 0.599 | 0.217[***] | 0.576[***] | 0.693[***] | -0.428[***] | **0.774** | |
| BI | 0.626-0.839 | 0.704 | 0.548 | 0.401[***] | 0.441[***] | 0.573[***] | -0.355[***] | 0.676[***] | **0.740** |

Notes: the bold fonts in the leading diagonals are the square root of AVEs. Off-diagonal elements are correlations among constructs. PSU, Perceived susceptibility; PSE, Perceived severity; PBE, Perceived benefits; PBA, Perceived barriers; SE, Self-efficient; BI, Behavior intention; CR, composite reliability; AVE, average variance extracted.

[***]$p < 0.001$.

## 3.3 Structural Model

SEM showed (Fig 2) that self-efficacy ($\beta = 0.482$, $p < 0.001$), perceived susceptibility ($\beta = 0.312$, $p < 0.001$), and perceived barriers ($\beta = -0.114$, $p = 0.024$) had a direct effect on behavioral intention, and perceived benefits ($\beta = 0.642$, $p < 0.001$) and perceived barriers ($\beta = -0.289$, $p < 0.001$) indirectly affected self-efficacy. Contrary to the hypothetical model, perceived benefits ($\beta = 0.241$, $p = 0.061$) and perceived severity ($\beta = -0.113$, $p = 0.293$) had no direct effect on behavioral intention. The final model explained 58.0% of the variance of HCV testing behavioral intention in MSM, indicating that the model had sufficient predictive utility.

The mediating effect was tested using the Bootstrap approach, with findings presented in Table 4. The study found that perceived benefits had an indirect influence on self-efficacy ($\beta = 0.309$, $p < 0.001$, 95%CI = 0.221 ~ 0.438). Perceived barriers had an indirect impact on self-efficacy ($\beta = -0.139$, $p < 0.001$, 95%CI = -0.214 ~ -0.090).

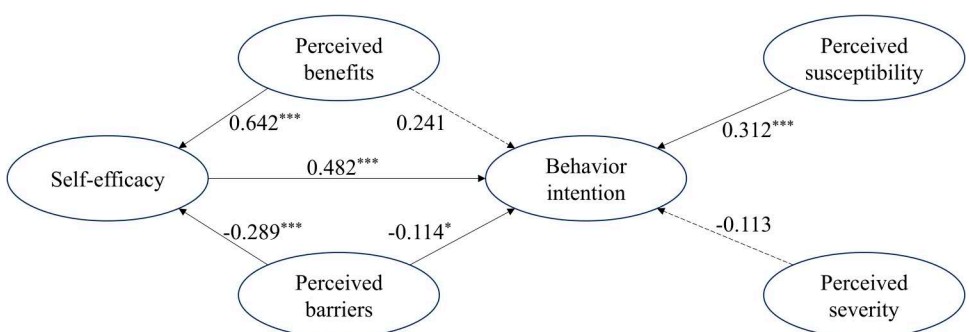

**Fig 2. The path diagram of structural model. (WLSMV was used for parameter estimation. The dashed line indicates that the coefficient is not statistically significant. [*]$p < 0.05$, [***]$p < 0.001$.).**

**Table 4. Standardized direct, indirect, and total effects of variables on behavioral intention.**

| Variable | Standardized Direct Effect | Standardized Indirect Effect | Standardized Total Effect |
|---|---|---|---|
| Perceived benefits | 0.241 (-0.037 ~ 0.483) | 0.309 (0.221 ~ 0.438) | 0.550 (0.325 ~ 0.753) |
| Perceived barriers | -0.114 (-0.211 ~ -0.005) | -0.139 (-0.214 ~ -0.090) | -0.254 (-0.346 ~ -0.167) |

## 4 Discussion

MSM are a population with greater likelihood of HCV, but the active test of MSM is poor. HCV infection is mostly occult, and early identification of infected individuals is important for the treatment and prevention of further transmit of the disease [27]. Investigating the influencing elements of HCV testing willingness is essential to raising the MSM HCV testing rate.

The results showed that 55.7% of respondents had been ever tested for HCV, higher than the 2017 survey result (41.0%) [15]. 90.9% of respondents were willing or very willing to take an HCV test in the future, and 71.7% said they were highly or very likely to take an HCV test in the next six months. Consistent with previous research results, MSM have a high willingness to test for HCV [28]. MSM's intention to test for HCV was positively influenced by perceived susceptibility, benefits, and self-efficacy, according to SEM analysis. Perceived barriers had a negative effect, while perceived severity had no significant effect, which was consistent with the research conclusions of female cervical cancer screening [29]. These factors explained 58.0% of the intention to test for HCV among MSM. Self-efficacy and perceived susceptibility are the most important factors in determining behavioral intention.

The study confirms that self-efficacy positively affects behavioral intention, which accords with prior research findings [30,31]. People with high self-efficacy mean they have higher confidence in completing the test and therefore have stronger behavioral intentions to test for HCV. In addition, consistent with previous studies, self-efficacy acted as a mediator between perceived benefits or obstacles and behavioral intentions [20,21]. Enhancing self-efficacy can significantly raise behavioral intention for HCV testing among MSM, and it is facilitated by raising awareness of the advantages of test and removing obstacles.

We found that behavioral intention was directly and positively influenced by perceived susceptibility. If MSM believe that they are more susceptible to HCV infection, they are more likely to test to determine whether they are infected, and their willingness to test for HCV will be higher. Previous studies have shown that the reasons for unwillingness to test for HCV often include "no risk factors" and "low risk of conscious infection" [32,33]. The lack of understanding of HCV will lead to a lower risk of self-perceived infection [32]. Therefore, when providing HCV-related knowledge to MSM, relevant organizations should emphasize the route of HCV transmission and high-risk behaviors, so that MSM can correctly understand their risk of HCV infection.

The study found no substantial influence of perceived severity on behavioral intention, which is similar to the results of other similar studies [29,34,35]. In addition, a review also found that the predictive effect of perceived severity on behavior in HBM is weak [36]. It may be that the public is more aware of the severity of HCV and believes that HCV infection can cause adverse health effects, resulting in a slight effect of perceived severity on desire to test. Therefore, the perceived severity cannot be used to identify the difference in the intention of MSM to detect HCV.

Our study showed that perceived benefits positively influenced self-efficacy and behavioral intention. Although there was no significant direct association between perceived advantages and behavioral intention, there was a strong link between perceived benefits and self-efficacy.

These results have also been observed in earlier research [19]. The mediating effect test showed that self-efficacy played a mediating role. Therefore, the self-efficacy and behavioral intention of HCV testing among MSM can be raised by publicizing the benefits of HCV testing. In addition, the initial phases of HCV infection is not easy to detect, and often the disease has progressed to a severe stage by the time it is found [37]. Therefore, relevant organizations should emphasize the benefits of early detection of HCV infection in disease treatment and prognosis to improve the perceived benefits of MSM.

Perceived barriers had a negative effect on behavioral intention in this study. Perceived barriers included "lack of information about HCV testing", "perceived discomfort of testing", "lack of time to test", and "high cost of testing", which were associated with decreased willingness to test for HCV among MSM. Similar results were found in previous studies [32, 33]. Self-efficacy mediated the effect of perceived barriers on behavioral intentions. When MSM perceived more barriers to HCV testing, this also reduced their confidence in their ability to complete HCV testing. Therefore, it is very important to take measures to eliminate the barriers in the process of HCV testing to improve the willingness of MSM to test.

Based on the above findings, health education on hepatitis C and HCV testing is recommended to increase risk awareness among MSM. In addition, efforts should be made to emphasize the benefits of HCV testing while reducing perceived barriers to HCV testing through improvements in external conditions, thereby enhancing self-efficacy and promoting active HCV screening in this population.

The present study has some limitations. This study employs non-probability sampling to reach the marginalized MSM community, which may result in biased results. Whether the proposed model can be generalized to other samples needs further verification. Second, the cross-sectional approach reduces our capacity to infer causal conclusions. Longitudinal follow-up investigations are needed to confirm the causal mechanism.

## 5  Conclusions

In China, MSM have a high willingness to test for HCV. Applying the SEM, we found that perceived susceptibility, benefits, barriers and self-efficacy are the predictors of behavioral intention. In addition, self-efficacy is an intermediary factor between perceived benefits or barriers to behavioral intention. Therefore, we should carry out targeted health education activities, publicize the positive significance of HCV testing, improve the accessibility of testing services, reduce existing barriers, and enhance the self - efficacy, so as to promote the practice of HCV testing.

## Supporting information

**S1 File.   Raw Research Data with Identifying Information Removed.**
(XLSX)

## Acknowledgements

Thanks to Guiqian Shi and Bin Lin for comments and suggestions throughout the investigation. We also thank all participants and investigators for their help.

## Author contributions

**Data curation:** Jiayan Li, Jing He, Guohui Yang.

**Investigation:** Jiayan Li, Jing He, Guohui Yang.

**Methodology:** Jiayan Li, Zhen Cao.

**Supervision:** Xiaoni Zhong.

**Writing – original draft:** Jiayan Li.

**Writing – review & editing:** Xiaoni Zhong.

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
