## [Decision Letter · Decision Letter 0]

7 Jan 2025

PONE-D-24-46278Influencing Factors of HCV testing Willingness among Men Who Have Sex with Men in China: A Structural Equation Modeling AnalysisPLOS ONE

Dear Dr. Zhong,

Thank you for submitting your manuscript to PLOS ONE. After careful consideration, we feel that it has merit but does not fully meet PLOS ONE’s publication criteria as it currently stands. Therefore, we invite you to submit a revised version of the manuscript that addresses the points raised during the review process.

Your manuscript was reviewed by two experts in the field. The reviewers identified many important problems in your submission and provided copious comments. Please carefully consider the attached comments and provide point-by-point responses. 

We look forward to receiving your revised manuscript.

Kind regards,

Yury E Khudyakov, PhD

Academic Editor

PLOS ONE

Journal Requirements:

This work was supported by the National Key Project for Infectious Diseases of the

Ministry of Science and Technology of China (2018ZX10721102-005). 

4. Ethics statement only appears at the end of the manuscript:

Your ethics statement should only appear in the Methods section of your manuscript. If your ethics statement is written in any section besides the Methods, please move it to the Methods section and delete it from any other section. Please ensure that your ethics statement is included in your manuscript, as the ethics statement entered into the online submission form will not be published alongside your manuscript. 

5. We note that you have indicated that there are restrictions to data sharing for this study. For studies involving human research participant data or other sensitive data, we encourage authors to share de-identified or anonymized data. However, when data cannot be publicly shared for ethical reasons, we allow authors to make their data sets available upon request. For information on unacceptable data access restrictions, please see http://journals.plos.org/plosone/s/data-availability#loc-unacceptable-data-access-restrictions. 

6. Please include a separate caption for each figure in your manuscript.

7. Please include your tables as part of your main manuscript and remove the individual files. Please note that supplementary tables (should remain/ be uploaded) as separate ""supporting information"" files.

Reviewers' comments:

Reviewer's Responses to Questions

**Comments to the Author**

1. Is the manuscript technically sound, and do the data support the conclusions?

Reviewer #1: Yes

Reviewer #2: Partly

2. Has the statistical analysis been performed appropriately and rigorously? 

Reviewer #1: Yes

Reviewer #2: Yes

3. Have the authors made all data underlying the findings in their manuscript fully available?

Reviewer #1: No

Reviewer #2: Yes

4. Is the manuscript presented in an intelligible fashion and written in standard English?

Reviewer #1: No

Reviewer #2: Yes

5. Review Comments to the Author

Reviewer #1: The authors have made a great effort in emphasising the willingness to test for HCV among MSM in China. I agree with the authors that the Health Belief Model can be of special interest when assessing the willingness to test. I do have, however a few concerns, which I will outline below. These concerns are mainly focused on the used language (not people first) and the manuscript can use some more elaboration on the studied topics and variables collected in the Method section.

Points in general:

Please make sure to use fair, accurate and respectful language and use people first language (See NIAID HIV Language Guide and People First Charter) throughout the manuscript. Some of the words or combination of words the authors use can be stigmatizing. This is very important.

Abstract:

- Line 23: I am puzzled by the use of detection rate, and wonder whether the authors actually mean testing rate? Detection rate relates to the number of positive HCV tests among those that were tested for HCV. Given your aim to assess influencing factors of HCV testing willingness, you might want to use testing rate instead of detection rate.

Introduction:

- Line 65-66: Do the authors mean anti-HCV antibody prevalence or RNA prevalence?

- Line 69: Had ever or recently undergone HCV testing? Please be more specific here.

- Also see my previous comment related to the background section: I am questioned whether you actually mean detection rate or testing rate? The 41% you are referring to is a testing rate.

Methods:

- Line 84-89: The in- and exclusion criteria are not entirely clear to me. Why do the authors exclude those with a significant mental disease or intellectual disability, and how do the authors determine that? And why exclude those who took fewer than three minutes to complete the questionnaire?

- Line 96-99: This is part of the results section, and is preferably mentioned at the very start of the results section.

- Line 114-116: It would be very helpful if the authors elaborate a bit more on the 23-item scale. Initially, it is not clear what the 23-item scale helps to assess and what it is for. And as the authors mention that the scale was based on the HBM and previous research, I also miss a reference. In the discussion part, the authors explain the definitions of the categories, but I would suggest to mention that in the Method section as this information is critical to understand the work.

- Could you please elaborate a bit more on the reference periods you use in the questionnaire? Is it ever tested for HCV, or in the past six months? And also do this for other variables. This is important to understand your results and translate them to current practice.

- What are the current HCV testing recommendation among MSM in China? This information would be much needed to understand the results of this work and to translate it to any implications. Is a testing rate of 55% really that low?

- And I wonder whether the authors also collected data on HIV status, as nothing is mentioned about that. We know from previous literature that new HCV infections are mostly found among MSM with HIV and MSM using HIV PrEP. Ofcourse, the epidemiological context of HCV might be different in China compared to Europe and the Amercas, but since MSM are considered a key population in China as well, this might also be the case there.

- The Method section could be extended a bit more. I am missing some information about the variables that were collected in the study, and the definitions that were used. For instance, looking at the method sectoin, it remains unclear that the authors collected data on STD and what type of STDs were included in this definition (all type of STDs, or only chalmydia, gonorrhea and syphilis?). I would suggest to mention this in the method section. For instance: We collected self-reported data on whether participants were diagnosed with any sexually transmitted infection, including XX, XX and XX. I also miss the reference period, is it ever diagnosed with an STD, or diagnosed in the past six months?

Results

- Line 155: Had been tested for HCV in the past six month, or ever? Again, reference periods are unclear.

- Line 155-156: Willing to be tested for HCV in the future, or in the upcoming six month? Reference periods are unclear.

Discussion

- Line 185: Here the authors talk about testing rate, make sure to be consistent throughout the paper using testing rate.

- Line 189-190: Higher willingness to test for HCV compared to? Or just a high willingness to test for HCV?

- In the discussion, the authors explain what is meant with self-efficacy, perceived susceptibility, perceived severity, perceived advantage. This type of information belongs to the method section, as it helps to understand the results.

- 55% of the sampled population reported being tested for HCV. Is that really that bad when almost two thirds of the sampled population only had one sexual partner, and thus might not be susceptible for HCV acquisition? I agree with the author’s limitation that the results may not be generalizable to other groups of MSM, but 55% might not even be that bad? But again, I don’t know the reference period of the question and the current HCV testing guidelines in China.

Reviewer #2: Dear Author,

Congratulations for choosing a pertinent topic for research and publication.

Find a few suggestions for further revisions and improvements in the written manuscript:

The author could add specific objectives of the study

In methodology the author could briefly explain the process of authentication of the recruited participants as it was an online recruitment. Author can clarify were the participants aware of the reward of 5 RMB (about $0.69) before giving consent or not. If yes, it can create ethical issues, which can be explained.

Likewise, author could briefly explain the meaning of problems in logical check and the way to identify it.

In the Table 2 legend: Characteristics of the study sample (N=857), however in the table N(n=680), which is not matching, and clarified by the author.

Conclusion could be aligned to aim & objectives of the study, the author could suggest recommendations according to the study results.

Regards

6. PLOS authors have the option to publish the peer review history of their article (what does this mean? ). If published, this will include your full peer review and any attached files.

**Do you want your identity to be public for this peer review?** For information about this choice, including consent withdrawal, please see our Privacy Policy .

Reviewer #1: No

Reviewer #2: **Yes: ** Dr Jarina Begum

---

## [Decision Letter · Decision Letter 1]

7 Mar 2025

Influencing Factors of HCV testing Willingness among Men Who Have Sex with Men in China: A Structural Equation Modeling Analysis

PONE-D-24-46278R1

Dear Dr. Zhong,

We’re pleased to inform you that your manuscript has been judged scientifically suitable for publication and will be formally accepted for publication once it meets all outstanding technical requirements.

Kind regards,

Yury E Khudyakov, PhD

Academic Editor

PLOS ONE

Additional Editor Comments (optional):

Reviewers' comments:

Reviewer's Responses to Questions

**Comments to the Author**

1. If the authors have adequately addressed your comments raised in a previous round of review and you feel that this manuscript is now acceptable for publication, you may indicate that here to bypass the “Comments to the Author” section, enter your conflict of interest statement in the “Confidential to Editor” section, and submit your "Accept" recommendation.

Reviewer #1: All comments have been addressed

Reviewer #2: All comments have been addressed

2. Is the manuscript technically sound, and do the data support the conclusions?

Reviewer #1: Yes

Reviewer #2: Yes

3. Has the statistical analysis been performed appropriately and rigorously? 

Reviewer #1: I Don't Know

Reviewer #2: Yes

4. Have the authors made all data underlying the findings in their manuscript fully available?

Reviewer #1: No

Reviewer #2: No

5. Is the manuscript presented in an intelligible fashion and written in standard English?

Reviewer #1: Yes

Reviewer #2: Yes

6. Review Comments to the Author

Reviewer #1: (No Response)

Reviewer #2: Congratulations to the author, for the revised manuscript. However, giving money to get responses could bring subjective bias and ethical concerns, thus author is recommended to avoid highlighting the specific statements in the article.

7. PLOS authors have the option to publish the peer review history of their article (what does this mean? ). If published, this will include your full peer review and any attached files.

**Do you want your identity to be public for this peer review?** For information about this choice, including consent withdrawal, please see our Privacy Policy .

Reviewer #1: No

Reviewer #2: **Yes: ** Dr Jarina Begum

---

## [Editor Report · Acceptance letter]

PONE-D-24-46278R1

PLOS ONE

Dear Dr. Zhong,

I'm pleased to inform you that your manuscript has been deemed suitable for publication in PLOS ONE. Congratulations! Your manuscript is now being handed over to our production team.

Kind regards,

on behalf of

Dr. Yury E Khudyakov

Academic Editor

PLOS ONE